# Biostimulants Using Humic Substances and Plant-Growth-Promoting Bacteria: Effects on Cassava (*Manihot esculentus*) and Okra (*Abelmoschus esculentus*) Yield

Luciano P. Canellas [1,*], Natália O. A. Canellas [1], Rakiely M. da Silva [1], Riccardo Spaccini [2,*], Gabriela Petroceli Mota [1] and Fábio L. Olivares [1,*]

1 Núcleo de Desenvolvimento de Insumos Biológicos para a Agricultura (NUDIBA), Universidade Estadual do Norte Fluminense Darcy Ribeiro (UENF), Campos dos Goytacazes, Rio de Janeiro 28013-602, Brazil
2 Centro Interdipartimentale di Ricerca CERMANU, Università di Napoli Federico II, Via Università 100, 80055 Portici, Italy
* Correspondence: canellas@uenf.br (L.P.C.); riccardo.spaccini@unina.it (R.S.); fabioliv@uenf.br (F.L.O.);
Tel.: +55-22-27397198 (L.P.C.)

**Abstract:** Traditional agriculture represents the most-extensive food-producing segment in the world. However, these agroecosystems are widely and closely associated with rural poverty, reflecting the dualism between the subsistence and the commodity-producing sector in the peripheric countries. Therefore, socially adapted technologies may be a reliable and helpful methodology to enhance subsistence crop production. Humic substances are natural organic biostimulants extractable as water suspensions from renewable sources such as agricultural biomass and farming residues. These easy-to-handle extracts may be mixed with plant-growth-promoting bacteria (PGPB) and used as biostimulants within a low-cost technological application in the circular economy strategy. Few investigations have been focused on the use of biostimulant practices on marginal or subsistence crops. Cassava (Manihot esculenta Crantz) and okra (Abelmoschus esculentus) are two essential foods for poor communities of rural territories in tropical and subtropical countries. The aim of this study was to evaluate the effect of the foliar application of a humic/PGPB mixed biostimulant on cassava and okra crops grown in an agricultural soil with very low natural fertility. In pot trials, the applied biostimulant improved the plant development with a 200% increase of the root weight in cassava, while the preservation of active diazotrophic bacteria was improved by 10- and 100-times in cassava and okra in the mixed formulation with humic acid. In real field systems, the plant treatment increased the yield of cassava and okra by 70% and 50%, respectively thereby allowing a simultaneous nitrogen savings with the best yield performance obtained at the lower N fertilization rate. The use of biostimulants can play a role in the transition process, helping the food security and the autonomy of impoverished farmers. Combining the elements of traditional knowledge and modern science is essential to create innovative technologies enabling the sustainable management of agroecosystems.

**Keywords:** physiological effects of humic acids; endophytic diazotrophic bacteria; vermicompost; rural poverty; traditional farming; food sovereignty; agricultural science and technology

## 1. Introduction

The epidemic crisis known as COVID-19 exposed the social vulnerability and difficulty of food accessibility for the poorest populations. Hunger affects 22% of the family farms in Brazil, and food scarcity and the diffusion of hunger in the countryside are relatively worse than in the urban peripheries [1]. The development and perpetuation of a functional dualism between the subsistence or sustainable production and commodity-producing sectors are objective outcomes of the laws of capital accumulation [2]. The expansion of modern science-driven farming turned Brazil into a global agricultural and agro-industrial

power [3] without enhancing food availability and contributing to the worsening of structural inequalities [4]. Furthermore, the supposed benefits of modern technologies have not prevented the advance of the degradation of native biomes in different Brazilian ecosystems. Accordingly, we lost more than 90% of the Atlantic Forest, 54% of the *Caatinga* (semi-arid native vegetation), 55% of the *Cerrado* (savannah-like biome), 20% of the Amazon, and 54% of the *Pampa* (native grasslands in the south of America). Most of these losses occurred in the last 30 years due to the indiscriminate increase and extension of intensive agricultural and agro-industrial activities [5,6].

Notwithstanding the extension of the agro-industrial sector, a significant portion of agricultural production comes from smallholders and small plots spread out in decentralized marginal environments [7]. This organization of farming is usually characterized by a valuable inherent resilience due to the limited use of external inputs, the coexistence of simultaneous multi-crop systems, and the centrality of qualified work. In more classical analytical terms, the low degree of commodification results in rural communities being faced with unpredictable and harsh problems and lifetime upsets during economic and socially turbulent periods [8]. It is significant that territorial markets became a focus of contestation and struggle during the first months of the brutal COVID-19 crisis. In the north of Rio de Janeiro State, the territorial market for customers' home-delivery rapidly expanded with e-commerce through mobile phone apps used by impoverished farmers, thus representing a viable alternative to the hegemony of the agro-food companies and related value chains. In addition, smallholders were faced with the need to rapidly increase crop production to meet these new demands, highlighting a non-simplistic relationship between the impoverished smallholder economy and a rapidly evolving non-conventional offsetting marketing strategy.

For many Brazilian rural territories, cassava (*Manihot esculenta Crantz*) and okra (*Abelmoschus esculentus*) are two essential foods for impoverished and smallholder communities. Cassava has an Amerindian origin and currently represents one of the main staple foods. Okra plants arrived in Brazil with slave ships, being incorporated into the primary diet of peripheral populations with two popular dishes: "*caruru*" (okra and dried shrimp in the north and northern areas of Brazil) and okra and chicken (in the southeast of Brazil) [9]. In other places, the fruits are boiled or fried and eaten as a vegetable or in a soup or dried form [10]. They are essentially subsistence crops grown with a low technological level, according to the green revolution (GR) concept, being however the pillar references for food and nutrition security. In addition, okra provides a good source of essential nutrients such as vitamins (C, A, B6, K, pyridoxine, folates), carotenoids, and flavonoids [11] and has antidiabetic properties [12,13]. Cassava roots are rich in starch and contain significant amounts of calcium, phosphorus, and vitamin C with a low production cost and tolerance to water deficit and low soil fertility [14]. In addition, cassava can be stored underground for a long time and harvested when necessary or convenient. However, despite the importance of both crops in feeding populations in underdeveloped countries, they can be considered neglected crops [15–17].

Socially adapted technologies can significantly increase food production in marginal areas under specific circumstances of low soil fertility and economics or environmentally untenable use of external inputs (mineral fertilizers, agrochemicals, etc.). The use of biostimulants based on humic substances (HSs) and plant-growth-promoting bacteria (PGPB) can be considered a reliable and easily affordable technical solution [18,19]. The physiological effects triggered by HSs [20–22] coupled with biological nitrogen fixation, phosphorus solubilization, and the production of plant regulatory substances by beneficial microorganisms [19] can further reduce the gaps in subsistence farming due to limiting factors such as water, nutrient availability, and biotic and abiotic stresses.

Altieri and Toledo [7] claimed that a few fundamental principles underlie the sustainability of traditional agricultural systems: species diversity, organic matter accumulation, the enhanced recycling of biomass and nutrients, the minimization of resource losses through soil cover and water harvesting, and the maintenance of high levels of functional

biodiversity. The use of biostimulants should be included as an auxiliary technology for plant adaptation, providing the possibility for farmers to have direct self-production and crop management. However, the agroecological transition processes are slow and delicate and require technologies and technical support that are not widely available. The main worldwide research activities are focused on the effect of commercial bioactive products for high-value crops, while less attention is devoted to the feasibility of technologies to support local farmers in transitioning from subsistence agriculture to a locally marketable level.

This work aimed to evaluate the application of low-cost technology based on a foliar spray of humic acids isolated from recycled biomasses mixed with beneficial bacteria consortia on cassava and okra cropping systems. The results for crop yields and nitrogen use efficiency are thoroughly discussed, considering the importance of socially adapted technologies.

## 2. Materials and Methods

### 2.1. Plant-Growth-Promoting Bacteria PGPB

The bacteria strains belong to the Bacteria Culture Collection of Laboratório de Biologia Celular e Tecidual (LBCT-UENF). All the bacteria species (*Herbaspirillum seropedicae* strain HRC54, *Burkholderia silvatlantica* strain UENF 117111, and *Burkholderia* sp. strain UENF 114111) were activated from the stock by growing the pre-inoculum in 5 mL of Digy's liquid medium in a rotatory shaker at 30 °C and 150 rpm for 48 h. Afterwards, a 50 μL suspension was transferred to a 250 mL flask containing 75 mL of the same liquid medium and growth conditions described herein. The final individual bacterium density was adjusted to $10^9$ cells $mL^{-1}$. Bacteria quantification associated with plants was performed using a semi-solid medium and the most-probable number (MNP) technique [23]). *Herbaspirillum seropedicae* strain HRC54 was counted by positive growth as pellicles formed in vials with JNFb semi-solid medium. The composition of JNFb medium was per liter of the following: malic acid (5.0 g); $K_2HPO_4$ (0.6 g); $KH_2PO_4$ (1.8 g) $MgSO_4\bullet7H_2O$ (0.2 g); NaCl (0.1 g); $CaCl_2$ (0.02 g); 0.5% bromothymol blue in 0.2 N KOH (2 mL); vitamin solution (1 mL); micronutrient solution (2 mL); 1.64% Fe•EDTA solution (4 mL); and KOH (4.5 g). In 100 mL, the vitamin solution contained: biotin (10 mg) and pyridoxine-HCl (20 mg); 1 L of the micronutrient solution consisted of: $CuSO_4$ (0.4 g); $ZnSO_4$ $7H_2O$ (0.12 g); $H_3BO_3$ (1.4 g); $Na_2MoO_4\bullet2H_2O$ (1.0 g); and $MnSO_4\bullet H_2O$ (1.5 g). The pH was adjusted to 5.8, and 1.9 $gL^{-1}$ of agar was added. For both *Burkholderia* strains, the semi-solid medium used was JMV, whose composition per L was: mannitol (5.0 g); $K_2HPO_4$ (0.6 g); $KH_2PO_4$ (1.8 g) $MgSO_4\bullet7H_2O$ (0.2 g); NaCl (0.1 g); $CaCl_2$ (0.02 g); bromothymol blue (0.5% solution in 0.2 M KOH) 2 mL; iron ethylenediamine tetraacetic acid (Fe-EDTA) (solution 1.64%) 4 mL; and micronutrient solution and vitamins described herein and agar 1.6 g $L^{-1}$, pH between 5.0 and 5.4). Total heterotrophic bacteria were obtained as CFU. $mL^{-1}$ in nutrient broth (NB) solid medium.

### 2.2. Humic Acids

The vermicompost used for the extraction of humic acids was processed by earthworms *Eisenia fetida* for 90 days in a continuous flow system using cattle manure as the starting biomass. The main chemical characteristics were: pH: 6.5; humic acids content: 325 mg $L^{-1}$; N: 1.7 mg $L^{-1}$; P: 2.7 mg $L^{-1}$; K: 58 mg $L^{-1}$. Soluble humic substances were extracted with 0.1 M NaOH at a 1:20 solid-liquid ratio by mechanical shaking for 6 h. The HAs was separated with acidification to pH 2 using 6 M HCl, followed by centrifugation at $5000\times g$. HAs were dialyzed against deionized water using a 1000-Da cutoff membrane (Spectrapor, USA) and freeze-dried. The elemental composition of HAs was characterized using a CHN analyzer (Perkin-Elmer 1483; Perkin-Elmer, Norwalk, CT, USA). After being corrected for ash, the carbon and nitrogen contents were 45.7 and 2.8 g·$kg^{-1}$.

The molecular characterization of HAs was performed by solid-state $^{13}$CNMR spectroscopy) cross-polarization magic angle spinning ($^{13}$C CPMAS NMR) with a Bruker AVANCE 300 NMR spectrometer equipped with a 4 mm-wide bore MAS probe, operating

at a $^{13}$C resonating frequency of 75.475 MHz. Four-thousand scans were collected over an acquisition time of 25 ms and a recycle delay of 2.0 s. All the free induction decays (FIDs) were transformed by applying a 4K zero filling and a line broadening of 100 Hz.

For the interpretation of the $^{13}$C-CPMAS-NMR spectra, the overall chemical shift range was split into six regions, related to the main organic functional groups: 0–45 ppm (aliphatic-C), 45–60 ppm (methoxyl-C and N-alkyl-C), 60–110 ppm (O-alkyl-C), 110–145 ppm (aromatic-C), 145–160 ppm (O-aryl-C), 160– 190 ppm (carboxyl-C) [24,25]. The relative contribution of each specific functional group was estimated by relating the area intensity of the corresponding spectral interval (Aiabs) to the total spectral area (A0-190abs): Ai% = (Aiabs/A0-190abs) × 100, i = 0–45, 45–60, 60–110, 110–145, 145–160, 160–190 (MestreNova 6.2.0 software, Mestre-lab Research, 2010).

To summarize the molecular characteristics of the organic extracts, the following dimensionless structural indexes were calculated from the combination of the relative C distribution in the specific functional groups of the NMR spectra [24,25]:

- The hydrophobic index is the ratio of signal intensities found in chemical shift intervals for apolar alkyl and aromatic C components over those of hydrophilic C molecules:
(1) HB/HI = Σ[(0–45) + (45–60)/2 + (110–160)]/Σ[(45–60)/2 + (60–110) + (160–190)];
- The alkyl ratio determines the relative contribution of apolar versus polar components of aliphatic molecules:
(2) A/OA = (0–45)/(60–110)
- The aromaticity index compares the area assigned to aromatic compounds to that of alkyl fractions:
(3) ARM = Σ[(45–60)/2 + (110–145) + (145–160)]/Σ[ (0–45) + (45–60)/2 + (60–110)]
- The lignin ratio, relates the area of methoxyl-C+N-alkyl groups to that of O-aryl-C:
(4) LigR = (45–60)/(145–160)

The HB, A/OA, and ARM parameters have been extensively applied to estimate either the biochemical stability of NOM, as well as the relation of the structural properties with the biostimulant activities of organic extracts, while the LR is a useful indicator to discriminate between signals owing to lignin and other phenolic moieties (lower LigR) with respect to the prevalent inclusion of peptidic clusters (larger LigR) in the 45–60 ppm interval [24,25].

### 2.3. Inoculant Preparation

Each bacteria cell suspension prepared as described above was pelleted by centrifugation (4000× *g* for 15 min) and resuspended in sterilized water at cell densities of $10^9$ colony-forming units (CFU) mL$^{-1}$. Then, the inoculant was prepared by diluting 200 mL of bacterial consortium (66.6 mL of strain) in 800 mL of humic substances at pH 7.0 to produce a final concentration of 48 mg C per liter and a final bacteria concentration of $5 \times 10^8$ cells/mL.

### 2.4. Pot Assay Experiment

Cassava stalks from the "blackly" cultivar were obtained from farmers in Campos dos Goytacazes, north of Rio de Janeiro state, and okra seeds (var Santa Cruz 47) were purchased at the local market. Stalks were uniform with comparable weights, diameters, and lengths, and the seeds were sown in 2.0 L pots filled with a superficial layer (0–20 cm) of an Ulfisol with the following chemical characteristics (Embrapa 1997): pH (H$_2$O) = 4.6; TOC = 10.4 g/kg; N = 1.1 g/kg Al$^{3+}$ = 0.10 cmol $_c$ dm$^{-3}$ (titration against NaOH); Ca$^{2+}$ = 0.80 cmol $_c$ dm$^{-3}$ and Mg$^{2+}$ = 1.20 cmol $_c$ dm$^{-3}$ (titration against EDTA); P = 4.45 mg dm$^{-3}$ (Mehlich 1 extraction and colorimetric determination); K = 52.0 mg dm$^{-3}$ (Mehlich 1 extraction and flame photometric determination); cation exchange capacity (CEC) = 5.3 cmol $_c$ dm$^{-3}$; sand, silt, and clay content of 880, 109, and 83 g/dm$^3$, respectively. The pots were irrigated twice a week, and after two weeks of cassava and okra germination, the inoculant preparation described above was manually applied using 50 mL per pot.

After three and six weeks, the okra and cassava seedlings were collected to evaluate the diazotrophic bacteria population associated with the roots. First, 1 g of root tissue was washed with tap water and macerated with a mortar and pestle into 9 mL of saline solution (NaCl 0.85%) to obtain a $10^{-1}$ dilution. Then, serial dilution until $10^{-7}$ was performed by transferring 1 mL of a dilution to a subsequent 9 mL tube. For the dilution ($10^{-2}$ to $10^{-7}$), an aliquot of 100 μL was inoculated into the vial containing 5 mL of semi-solid medium, with three replicates per dilution. The JNFb semi-solid medium was used for *H. seropedicae*, and the JMV semi-solid medium was used for both *Burkholderia* strains for diazotrophic population estimation by the NMP method. The vials were incubated in a growth chamber at 30 °C for 7 d. A white pellicle indicated positive growth for diazotrophs in inoculated and control plants. The inoculated bacteria's identity was confirmed by cell shape under phase contrast microscopy and colony appearance in a solid medium.

### 2.5. Field Experiment

A field experiment was carried out on an Ultisol located in the Lagoa de Cima district of Campos dos Goytacazes, Rio de Janeiro, Brazil (21°46′19″S 41°30′56″W; altitude 14 m) (in the area where the soil sample was collected for the pot assay experiment). The main soil properties, the map of the site location, and the weather conditions are included in Table S1 and Figures S1 and S2 in the Supporting Materials. The trial was set up in a completely randomized design with four repetitions, using one application of humic acids isolated from vermicompost at 48 mg $CL^{-1}$, a suspension of bacteria consortia using *H. seropedicae* HRC54, *Burkholderia* sp. UENF 114111, and *B. silvatlantica* UENF 117111 ($2 \times 10^8$ cells $mL^{-1}$), and plants without biostimulant application (control treatment). We imposed four nitrogen levels 0, 20, 40, and 60 kg N from urea $ha^{-1}$ in cassava (var. blackly) and okra (var. Santa Cruz 47). For both (cassava and okra), the area was prepared by ploughing and harrowing, followed by the opening of furrows spaced 0.6 m for two lines of cassava and 1.0 m for okra lines. The experimental plots consisted of one and two lines of 5.00 m for okra and cassava, respectively, and the pits were spaced 0.5 m. Then, 5 L of cattle manure was added to the pit. The urea was added 30 and 45 days after okra and cassava germination. The treatments (biostimulant by foliar spray) were applied 5 days before nitrogen fertilization. The roots (cassava) and fruits (okra) were harvested for crop yield analysis. The data were analyzed using a regression model after ANOVA.

## 3. Results

### 3.1. Chemical Characteristics of Humic Acids Used as a Vehicle for PGPB

The CP-MAS $^{13}$C NMR spectrum of HAs was characterized by a predominance of signals associated with lignin and aromatic derivatives (Figure 1). The highest sharp resonance at 56 ppm is related to the methoxyl substituents linked to the aromatic ring of the guaiacyl and siringyl lignin structural units, as well as of the polyphenolic and lignan components. This intense peak may also include the possible contribution of C-N bonds in peptidic moieties [24]. The multiple resonances shown in the wide bands extended between the 110 and 145 ppm chemical shift region, further supporting the incorporation of different aromatic components, followed by the peaks at 147 and 152 ppm deriving from the O-bearing aryl-C of the lignin and phenolic molecules [24,25].

The aliphatic components of humic extract are represented by the signals of both apolar alkyl-C and polar O-alkyl-C in the 0–45 ppm and 60–110 ppm interval, respectively, The large peak at 30 ppm is related to the contiguous resonances of methylene groups in various long-chain compounds pertaining to either fatty acids and alcohols of plant and lipid waxes or to bio-polyester components such as alkyl dioic and hydroxyl carboxylic acids closely associated with lignin in the suberin layers of higher plants [25]. The smaller, but distinct peaks at 35–40 ppm may be assigned to the tertiary and quaternary C nucleus in the assembled ring of cyclic compounds such as sterol, flavonoids, and lignans. The polar O-alkyl-C fractions are made up by the pyrasonide units of carbohydrates and polysaccharides, identified by the overlapping frequencies of carbon nuclei in position

2,3,5 at 72 ppm and from the di-O-alkyl anomeric C at105/106 ppm. The lack of the specific shoulder of Carbon 4, involved in the glycosidic bonds of cellulose fibers, around 81–88 ppm suggests the preferential co-extraction of hemicellulose molecules associated with lignin in the plant cell walls, as well as the solubilization of unbound low-molecular-weight sugars [25]. Finally, the final band at 176 ppm indicates the inclusion of carbonyl groups pertaining to various compounds such as aliphatic acids, amino acid moieties, and side groups of acid functions of hemicellulose.

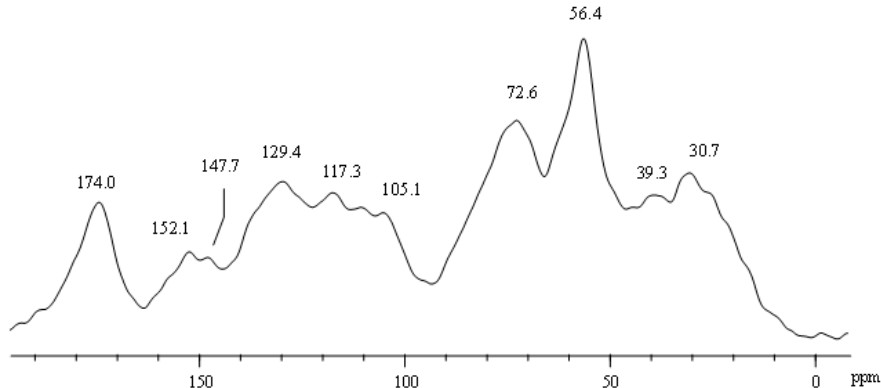

**Figure 1.** CP/MAS $^{13}$C NMR spectrum of humic acids isolated from cattle manure vermicompost.

The value close to unity obtained from the calculation of the HB dimensionless index (Table 1) highlights the almost fair repartition of functional groups between hydrophilic and hydrophobic compounds, while the comparable level of A/OA and Ar indicates the equal contribution of apolar aliphatic and aromatic molecules. The low values shown by the lignin ratio outline a close correspondence between the intensities of the NMR signals of methoxyl (45–60 ppm) and phenolic (145–160 ppm) regions, thus confirming the selective solubilization of lignin molecules [24,25]. These molecular properties were previously related to both the high biological activity and the large preservation of PGPB bacteria of humic acids from vermicomposts [26,27].

**Table 1.** Relative distribution (%) of signal areas over chemical shift regions (ppm) and structural indexes [a] in the CP/MAS $^{13}$C-NMR spectra of humic acids (HAs).

| | 190–160 C=O | 160–145 O-aryl-C | 145–110 Aryl-C | 110–60 O-Alkyl-C | 60–45 CH$_3$O/C-N | 45–0 Alkyl-C | HB | A/OA | Ar | LR |
|---|---|---|---|---|---|---|---|---|---|---|
| HAs | 9.5 | 5.7 | 20.1 | 31.0 | 14.1 | 19.5 | 1.1 | 0.6 | 0.6 | 2.5 |

[a] HB/HI = Σ[(0–45) + (45–60)/2 + (110–160)]/Σ[(45–60)/2 + (60–110)+ (160–190)]; A/OA= (0–45)/(60–110); Ar = Σ[(45–60)/2 +(110–145) + (145–160)]/Σ[(0–45) + (45–60)/2 +(60–110)]; LigR = (45–60)/(145–160).

### 3.2. Initial Plant Growth and Diazotrophic Bacteria Population (Pot Assay)

The applied bioactive mixed formulation increased the root and shoot fresh weight for both cassava and okra compared with non-inoculated plants (Figure 2). At 45 days, the best biostimulation performance was found in cassava seedlings treated with biostimulant, which showed an improvement of around 200% of the root weight (Figure 2A). Compared to control plants, the inoculation promoted also a significantly larger shoot weight in cassava and okra (Figure 2A, B), and an overall faster plant development (Figure 2C,D).

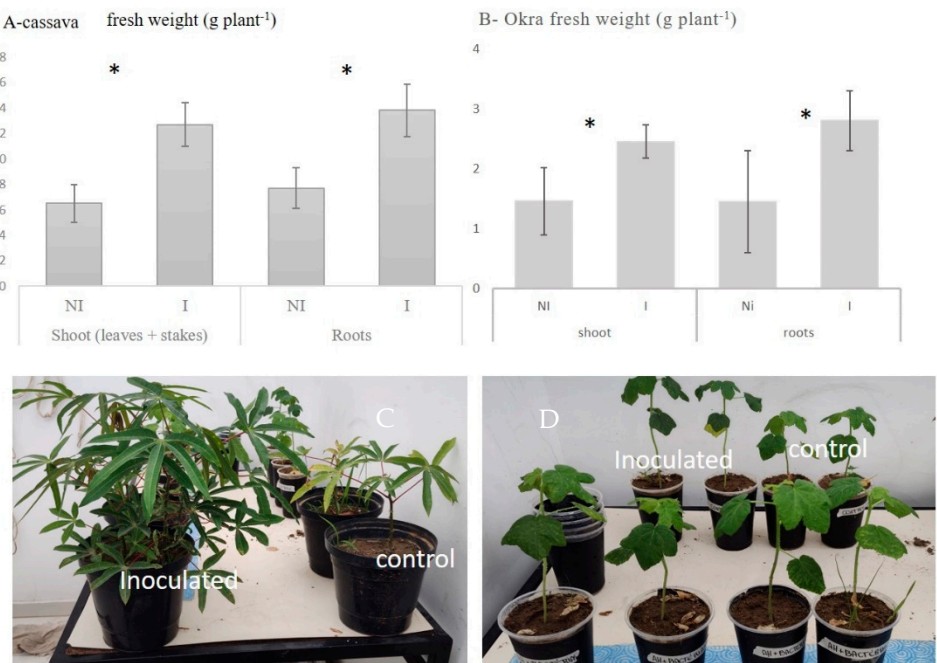

**Figure 2.** Root and shoot fresh weight (g plant$^{-1}$) of (**A**) cassava (90 days) and (**B**) okra (30 days). Seedling development: (**C**) cassava, (**D**) okra. Treatments: control plants (NI), inoculated (I): a consortium of *Herbaspirillum seropedicae* strain HRC54, *Burkholderia* sp. UENF 114111, and *B. silvatlantica* UENF 11711 plus 4 mM of C L$^{-1}$ humic acids isolated from vermicompost. The values represent the mean $\pm$ the standard deviation. * F-test significance ($p < 0.05$).

The total yield of heterotrophic bacteria estimated in the *NB* solid medium was similar between indigenous and inoculated communities associated with cassava and okra (Figure 3). However, for diazotrophic bacteria, we observed an increase of more than 10- and 100-times in the bacteria community grown in the JMV or JNFb medium related to the control in cassava roots. A similar trend was observed for okra roots, whose inoculated plants revealed a larger diazotrophic population, mainly in the JMV medium, while a less marked differences was found in the JNFb medium between biostimulant-treated plants related to the control (Figure 3).

It is worth mentioning that the indigenous diazotrophic bacteria associated with cassava and okra plants do not resemble the bacteria species that compose the biostimulant formulation (*H. seropedicae* and *Burkholderia* spp.), as highlighted by the phase contrast microscopy and bacteria colony phenotyping.

### 3.3. Field Experiment

The crop yield of cassava var. "*pretinha*" had an average increase of 42% following the biostimulant application (Figure 4). Furthermore, the difference between inoculated and control plants was even enhanced under a lower urea concentration.

Without N fertilization, the measured root fresh weight in biostimulant-treated parcels was 75% larger than the control reference. The highest production was obtained with the inoculated plants and 20 kg N urea ha$^{-1}$, which yielded 28.4 Mg ha$^{-1}$ of fresh root, which is almost two-fold the average Brazilian production set around 15 Mg ha$^{-1}$ according to EMBRAPA [28]. Following the increase of the urea fertilization from 20 to 40 kg ha$^{-1}$ or 60 kg ha$^{-1}$, the inoculated crops underwent a decreasing response, although maintaining a significant positive gradient with respect to the exclusive addition of mineral fertilizer (Figure 4). This finding may be related to the contrasting effect of high available N on the tuberous root weight due to the preferential priming of shoot and epigeal tissues [29]. These data suit the conventional farming management of cassava cultivation based on the limitation or the shortage of N fertilizer. In fact, 74.5% of cassava producers in the north

of Rio de Janeiro State do not use any fertilization considering a cluster of small fields (<2 ha) in rural marginal areas with low natural soil fertility [30]. The remarkable increase observed in crop yield and growth due to the inoculation of the beneficial bacterial and humic acids represents a significant gain in food availability and security coupled with the belowground storing and stabilized supply throughout the year.

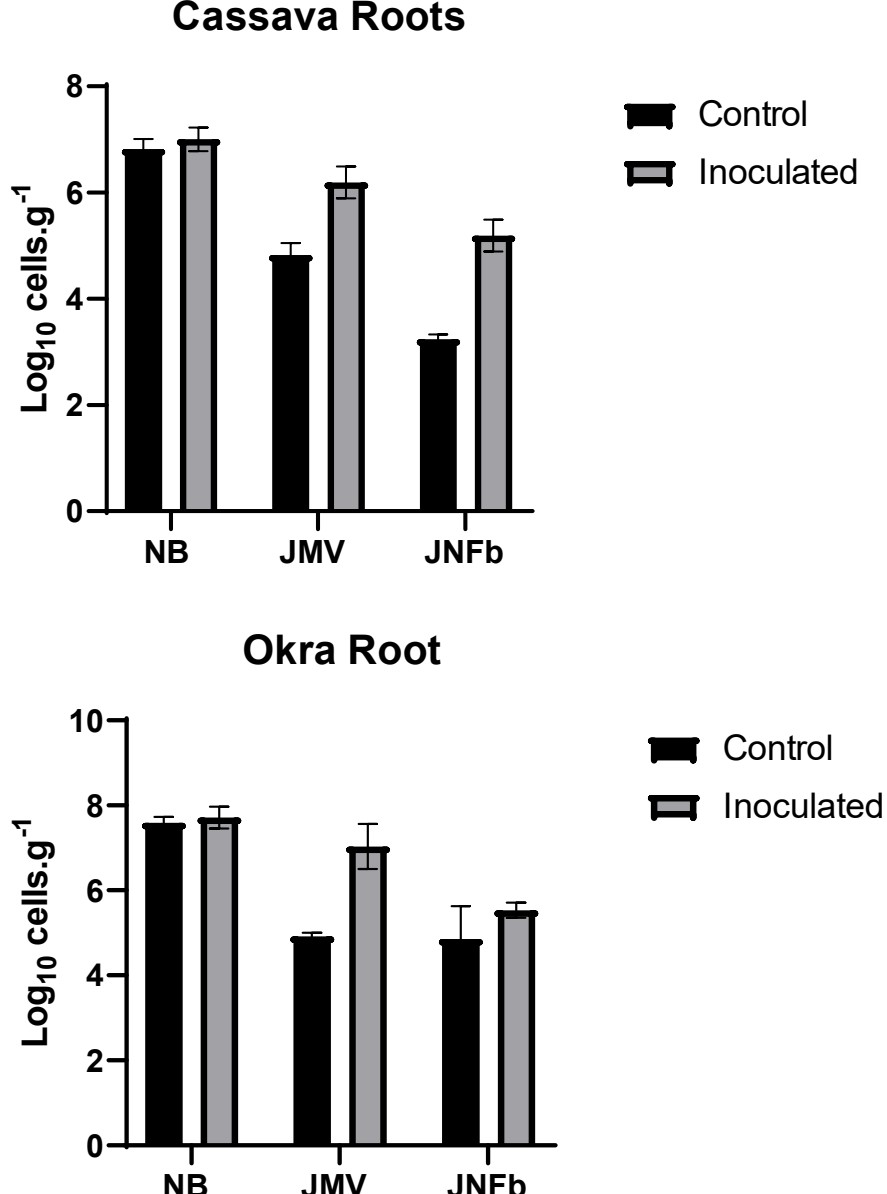

**Figure 3.** Bacteria quantification values (expressed as $\log_{10}$) associated with okra and cassava roots inoculated with a mixture of *Herbaspirillum seropedicae* strain HRC 54, *Burkholderia silvatlantica* UENF 117111, and *Burkholderia* sp. strain UENF 114111. Diazotrophic *Burkholderia* counts in JMV (C-mannitol) semi-solid medium and nitrogen-fixing *H. seropedicae* counts in JNFb (C-malic acid) semi-solid medium (n = 3). Total heterotrophic bacteria counts in nutrient broth (NB) solid medium (n = 4). The values represent the mean ± the standard deviation.

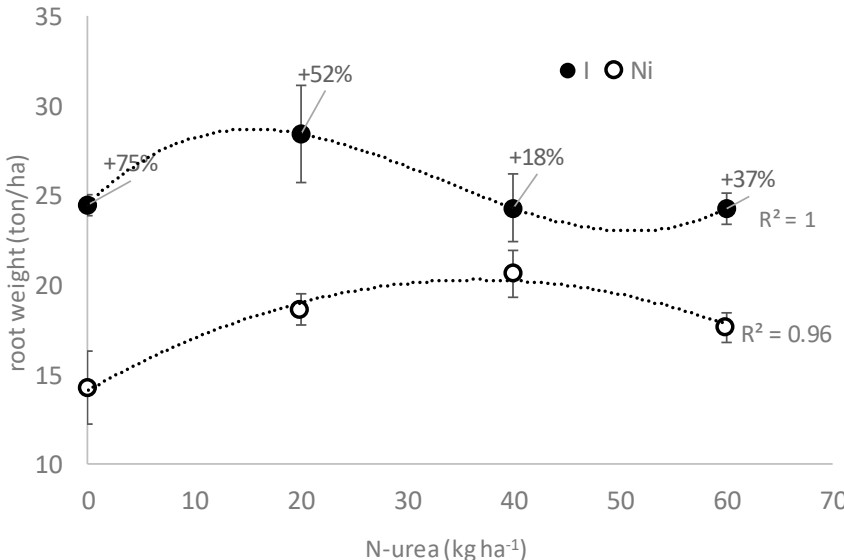

**Figure 4.** Root fresh matter yield of cassava var. "*pretinha*" inoculated (I = ●) or not (NI-○) with *H. seropedicae* HRC54, *Burkholderia* sp. 103, and *B. silvatlantica* 101 in the presence of humic acids (4 mM C L$^1$) at 60 and 90 days after germination. The values represent the mean followed by the SD (n = 4).

A comparable positive influence of the mixed biostimulant was even found for the okra fruit (Figure 5).

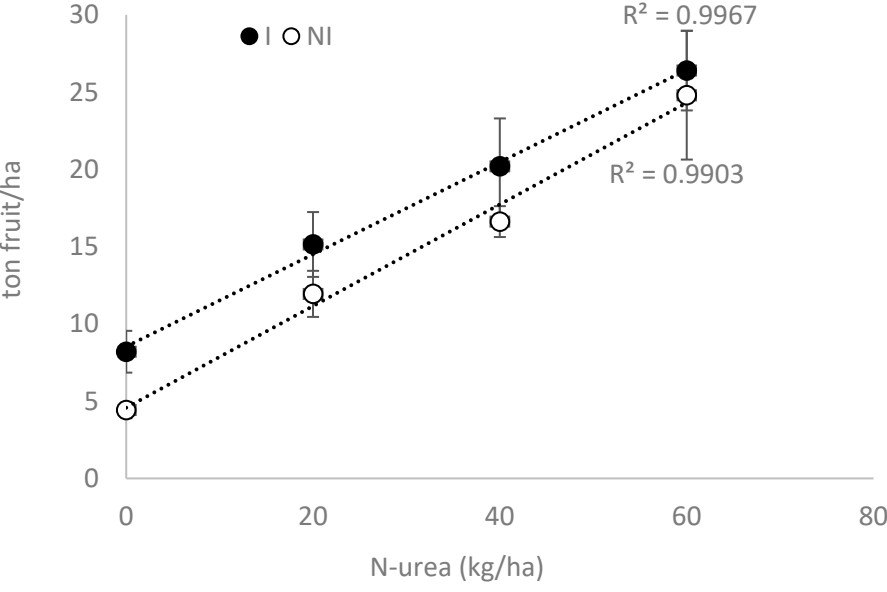

**Figure 5.** Fruit fresh matter yield of okra var. *Santa Cruz 47* inoculated (I = ●) or not (NI-○) with *H. seropedicae* HRC54, *Burkholderia* sp. 103, and *B. silvatlantica* 101 in the presence of humic acids (4 mM C L$^{-1}$) at 30 days after germination. The values represent the mean followed by the standard deviation (n = 4).

An average differential yield of 35% with respect to the control plants was observed with inoculation, ranging from the maximum increase of around 85% without urea to the lowest differences of +6% with 60 kg N ha$^{-1}$. Unlike the cassava roots, the fructification process of okra may profit from mineral fertilization [31]. In the present field trials, the yield of the control plants increased almost linearly with the nitrogen fertilization rate, thus retaining the potential benefits of inoculant application (Figure 5).

### 4. Discussion

The current strategies to reduce the yield gap in marginal agricultural areas include the use of plants from commercial breeding, combined with the associated larger resource inputs such as irrigation, fertilization, and the control of pests, diseases, and weeds with agrochemicals [14,30]. However, cassava is a very drought-tolerant and water-efficient crop, being also exceptionally adapted to high soil acidity and low levels of available phosphorus. In addition, there are no economically important pests or diseases in cassava in the Rio de Janeiro cultivation area , with a consequent limited need for treatments against biotic stresses. Locally available genotypes still dominate okra, such as open pollinated variety Santa Cruz 47, which combines high yield performance with low input strategies. Different from commodities, the price paid for okra pods is almost on the production cost threshold. In these cases, the consequences of GR's technologies for impoverished farmers can be economically and environmentally harmful. Since cassava and okra cannot be grown in temperate climates, the application of alternative management approaches has never received the same research interest dedicated to cereals, soybeans, and potatoes or horticultural crops in developed countries [32]. Research on cassava had been minimal until the early 1970s, when the international research centers—CIAT in Colombia, EMBRAPA Cruz das Almas in Brazil and IITA in Nigeria—received the mandate for cassava research and development. In Brazil, the okra research centers are limited (Instituto Agronômico de Campinas (IAC) and Embrapa Vegetables), while more scientific and technical investigation have been activated in Asia and Africa. Nevertheless, the number of researchers working on cassava and okra, and the research budgets dedicated to them, are minimal in comparison with those for most of the competing crops, this resulting in an evident and lasting yield gap [32].

During the pandemic crisis, it was possible to feel the social pressure to increase the production of agricultural goods and basic reference foods. Common sense pointed to the use of technologies not available on the market due to the spatial and temporal mismatching of production, transport, and supply chains (e.g., fertilizer shortage). However, this economic and social crisis was not a one-off case: basic food availability has stagnated or declined steadily in the recent past. For example, Brazil's cassava cultivation and production in the last 20 years has decreased in amount and diffusion area. From 1990 to 2020, the extension of planted fields was reduced by 38%, and output fell by 25%. According to official Brazilian data, the average productivity set at 12.5-ton ha$^{-1}$ in 1990 slowly went up to 15.2 tons ha$^{-1}$ in 2015 and stagnated at 15 tons ha$^{-1}$ afterwards [28]. The agricultural system dedicated to okra in Brazil has reached around 20,500 ha since 1990, being 95% locally developed and low-productivity open-pollinated cultivars seeds and only 5% hybrid cultivars. A similar framework is found for the world's largest okra producers such as India, Nigeria, Sudan, Mali, and Pakistan, which, notwithstanding the agricultural potential according to the indexing protocols of rating agencies, are not considered among the top developed countries. Furthermore, the productive gap suffered from the combination of technical deficit and the deterioration of soil fertility and natural resources, linked to inadequate or inappropriate intensive management practices [32]. The relapses of GR [33] go beyond environmental issues, including land and power concentration, inequality, and extreme poverty in rural areas, but remains a *"powerful legend which continues to reverberate, inspire, and influence perspectives and practices in agricultural science and technology"*, according to Cabral et al. [34]. Since the profit-based approaches and the resulting public opinion consider the alleged remoteness of poor and small farmers from modern technologies as the main reason for low yields and income [30], two issues will be hereby discussed: the meaning and often misleading concept of modernizing technologies for family-based agriculture and the role of scientifically grounded alternative support in the transition processes.

During the first months of the COVID-19 crisis, territorial markets became, almost everywhere in the world, the crucial local commercialization centers of poor farmers' production with a clear incentive to increase the output [8]. The chain of food intermediaries

was interrupted, and short circuits of direct marketing were encouraged with the dissemination of social apps. However, access to fertilizers, even at inflated prices, was scarce due to transport and exchange problems. In addition, the rising costs erode farmers' profits, counteracting and nullifying the benefits of higher food demands. In this context, the use of biological inputs to promote crop yield can be a helpful tool to match sustainable field management, consumers' requirements, and farmers' revenue. The results of pot and field experiments outlined that the smart application of mixed abiotic and biotic biostimulants promoted both the stimulation of cassava and okra in the initial growth phase and the preservation of both the population and biological activity of diazotrophic bacteria in the root and shoot fresh tissues (Figures 2 and 3). This initial performance resulted in a larger crop yield, and the effect was even emphasized under low nitrogen fertilization (Figures 4 and 5).

The beneficial properties of the mixed biostimulant rely on the synergistic combination of humic materials and PGPB. The bioactive effect of HAs is related to the contiguous presence of hydrophilic and hydrophobic domains, which trigger the self-aggregation into a micelle-like structure [35,36]. The conformational behavior promotes the suitable dissolution in water suspensions, thereby favoring a surface tension activity with a potentially significant adhesion capability on natural inorganic and organic materials [37,38]. Moreover, the structural characteristics allow the suitable incorporation and preservation of microorganisms and bioactive compounds such as phenolic and lignin components [24,25,27], acting as viable carriers of biostimulants on plant tissues [19,27,39].

The effects of humic substances on plant physiology have been a matter of extensive review [20–22,40,41]. Various enzymes have been identified as being involved in a plant response to HS action including proton pumps, glycolytic enzymes, the tricarboxylic acid cycle, and the shikimic acid pathway. A consensus was achieved concerning multiple regulating functions of HS, including direct stimulation of root growth and root hair proliferation, modulation of the release of protons and root exudates, regulation of ion uptake rates, redox reactions, and others. The biostimulant proprieties of humic substances pushed the market into a rising trend, expected to increase global returns at a rate between 9 and 13.4% by 2025 [42].

The corporate world has appropriated this natural component in a business that revolves around USD 1 billion today. This consists of mining peat and leonardite, destroying endangered wetland ecosystems worldwide. However, as demonstrated previously, the bioactivity of humic substances isolated from compost and vermicompost, a renewable source of stabilized organic matter, is as high or higher than peat or leonardite [43]. Vermicomposts generally have a finer structure, contain more nutrients, and have higher microbial activity than other types of composts. Extracts from vermicomposts are innovative biostimulants requiring relatively smaller quantities to cover a larger production area while extracting all the excellent biochemical properties of solid vermicomposts [44]. The farmers can conduct and control the entire process without corporate involvement. The inoculants using different plant-growth-promoting bacteria are available in the conventional market. Impoverished farmers can dilute them in low concentrations of humic substances before plant application.

The number of reports considering biofertilizers with diazotrophic bacteria in cassava and okra is scarce. Mal et al. [45] applied biofertilizers with *Azospirillum* and *Azotobacter* and *Azospirillum* and phosphorus-solubilizing bacteria together or not with NPK and manure. The biofertilizer allowed reducing the NPK, enhancing the nutrient use efficiency. Balota et al. [46] identified the presence of *Klebsiella* sp., *Azospirillum lipoferum*, and *Burkholderia* spp. naturally occurring in stalks and cassava roots.

A critical issue to be considered is that, traditionally, the mechanization of smallholder work has been low, as well as the dependence on fossil fuels. Therefore, any change that amplifies the demand for labor should be carefully considered. The ecological intensification of crop yield management is based on enhancing the soil organic matter and nutrient cycling, water efficiency uses, and biodiversity [47]. Changes in agricultural practices and increased

diversity in production involve multiple and interdependent adaptations to managing the whole agroecosystem [48]. Still, the first step is always to restore the soil's ability to promote life. Organic matter amendments have a central role in this process. However, increasing organic matter content in tropical soils is challenging, especially highly weathered ones. One of the most-efficient practices for increasing soil organic matter content in tropical and subtropical environments in both forest and agroecosystems is amendment with organic matter enriched with hydrophobic groups [49,50]. This option may require the availability of organic biomass's management and further transport and application steps, increasing the demand for labor. It is well known and reported that the lack of transport of manure and high labor demands are the most-cited factors leading to the non-adoption of alternative agriculture management [51]. However, a suitable soil organic matter management may be also obtained with low-input agricultural practices [52,53]. It is possible to obtain small vermicompost productions with organic residues from the home and agricultural by-products [54,55] with easy handling to extract soluble humic substances, which allows covering a relatively large area of cultivation because small concentrations are used. This approach is also qualified for the introduction of a gender-sensitive issue. Most agroecology assessments focus on ecological benefits, such as substituting chemical fertilizers and crop diversification, but little attention is given to the gender aspects. Vermicomposting is a typical low-cost technology for processing or converting organic waste into high-quality composts. It creates conditions for women's promotion, since they are almost always responsible for household waste and attention to small animals. The patriarchal heritage in the Brazilian countryside is a cultural chasm to be bridged for the achievement of equality, a basic agroecology assumption.

## 5. Conclusions

A smart foliar application of an easy-to-handle product containing beneficial bacteria and humic acids isolated from a renewable source of organic matter at low concentrations resulted in a significant yield increase in both cassava and okra yield combined with low fertilization requirements. For practical methodological approaches, it is worthwhile to highlight the feasibility of low-cost technological input. The application with a backpack sprayer is quick and easy and is part of every farmer's routine. Moreover, a water dispersion may be also conceived of for a furrow distribution, as well as for root treatment before transplanting. It is non-toxic and has no negative consequences for the environment. Being inserted within an on-farm context can increase the production and add high-level technology, and it is easy to implement, contributing to the autonomy, food security, and income of smallholders. The use of these mixed biostimulants can effectively support the crop productivity of plants cultivated by subsistence farmers that do not undergo genetic breeding programs.

There is a risk that this observed increase will be narrated in an epic way linked to a positivist epistemology celebratory of science or described as a successful case of replacing chemical inputs with organic inputs to reduce negative environmental impacts and raise the production limits of production-oriented agriculture. In both cases, it is essential to disrupt the interests of the sociotechnical imagination. It is possible to combine elements of traditional knowledge and modern science while being mindful of the hegemonic paradigm's power dynamics and the need to resist co-optation [56]. Finally, we reproduce an excerpt from the work of Giller et al. (2021): [57] *the food system is best viewed as an integral part of the much broader network of economic, social and political relations. It follows that many of the faults ascribed to the food system–including hunger, food poverty, poor labor relations, and corporate dominance–will not be successfully addressed by action within the food system but only through higher-level political and economic change*. However, at a local level, the subsistence farming of cassava and okra, in which the farmers control their genetic heritage, can benefit from biotechnological processes as long as they are managed (at all stages) by themselves. Bio-inputs can go beyond substitution agriculture: they can also question the technology production model.

**Supplementary Materials:** The following Supporting Information can be downloaded at: https://www.mdpi.com/article/10.3390/agronomy13010080/s1, Table S1 Main characteristics of soil samples used for pot and field trials; Figure S1: Location of experimental site; Figure S2 Annual weather conditions in the experimental area.

**Author Contributions:** Conceptualization, L.P.C. and F.L.O.; field and green house research: L.P.C. and N.O.A.C.; data analysis: R.M.d.S.; bacteria counting: G.P.M. spectral analysis: R.S.; writing—original draft preparation, L.P.C.; writing—review and editing, F.L.O and R.S. All authors have read and agreed to the published version of the manuscript.

**Funding:** This work was supported by Fundação Carlos Chagas Filho de Amparo à Pesquisa do Estado do Rio de Janeiro (FAPERJ) Cientista do Nosso Estado, Conselho Nacional de Desenvolvimento de Pesquisa e Tecnologia (CNPq).

**Data Availability Statement:** Not applicable.

**Acknowledgments:** Not applicable.

**Conflicts of Interest:** Not applicable.

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
