# Peer review of "Biostimulants Using Humic Substances and Plant-Growth-Promoting Bacteria: Effects on Cassava (Manihot esculentus) and Okra (Abelmoschus esculentus) Yield"

_agronomy, doi:10.3390/agronomy13010080_

Round 1

Reviewer 1 Report

Major corrections are required for the publication of this manuscript entitled “Biostimulants using humic substances and plant-growth promoting bacteria: effects on cassava (Manihot esculentus) and okra (Abelmoschus esculentus) yield”.

1.     Please rewrite the abstract and add how your study is better than others. Why were the results of other parameters, such as fresh root/shoot weight from the pot experiment and field, the bacterial community, and others not included? Authors only provided an abstract explanation of the yield parameters.

2.     Please improve the language of the manuscript.

3.     Improve the introduction section with latest references from last five years (2018-2022).

4.     Please include the standard error values/or a, b, c (significant difference) in figure 3.

5.     Please discuss the mechanism how the treated plants improve yield production and the outcome of your study and how it benefits for others. Update the discussion section with recent references related to your study.

6.     Please include the conclusion of this study.

Reviewer 2 Report

Dear Authors,

you have presented a paper in which you attempted to study the valued effect of foliar application of a mixed humic/PGPB biostimulant on the yield of cassava and okra grown on agricultural soil with very low natural fertility. In my opinion, the work has a very high value, both substantive and practical. The world market for biostimulants is growing every year, but it is necessary to look for more and more perfect substances, as close as possible to naturally occurring compounds. Please consider some of the following comments:

1. in the abstract section, state the location-where you conducted the research.

2. are you able to characterize the local market for biostimulants? Do they require registration in your legislation?

3. please number the formulas in the material and methods section

4. paste a map indicating the experimental location.

5. characterize the test site in detail- state the type of soils, weather conditions, etc. 

6. Need to develop an application-at least 3, the most important. 

Round 2

Reviewer 2 Report

I accept